# Enhancement of Vancomycin Potential against Pathogenic Bacterial Strains via Gold Nano-Formulations: A Nano-Antibiotic Approach

**DOI:** 10.3390/ma15031108

**Published:** 2022-01-31

**Authors:** Turki Al Hagbani, Hemant Yadav, Afrasim Moin, Amr Selim Abu Lila, Khalid Mehmood, Farhan Alshammari, Salman Khan, El-Sayed Khafagy, Talib Hussain, Syed Mohd Danish Rizvi, Marwa H. Abdallah

**Affiliations:** 1Department of Pharmaceutics, College of Pharmacy, University of Ha’il, Ha’il 81442, Saudi Arabia; t.alhagbani@uoh.edu.sa (T.A.H.); afrasimmoin@yahoo.co.in (A.M.); a.abulila@uoh.edu.sa (A.S.A.L.); frh.alshammari@uoh.edu.sa (F.A.); mh.abdallah@uoh.edu.sa (M.H.A.); 2Department of Pharmaceutics, RAK College of Pharmaceutical Sciences, RAK Medical & Health Sciences University, Ras Al Khaimah 11172, United Arab Emirates; hemant@rakmhsu.ac.ae; 3Department of Pharmaceutics and Industrial Pharmacy, Faculty of Pharmacy, Zagazig University, Zagazig 44519, Egypt; 4Department of Pharmacy, Abbottabad University of Science and Technology, Havelian 22010, Pakistan; adckhalid@gmail.com; 5Nanomedicine and Nanotechnology Lab, Department of Biosciences, Integral University, Lucknow 226026, India; salmank@iul.ac.in; 6Department of Pharmaceutics, College of Pharmacy, Prince Sattam Bin Abdulaziz University, Al-kharj 11942, Saudi Arabia; e.khafagy@psau.edu.sa; 7Department of Pharmaceutics and Industrial Pharmacy, Faculty of Pharmacy, Suez Canal University, Ismailia 41522, Egypt; 8Department of Pharmacology and Toxicology, College of Pharmacy, University of Ha’il, Ha’il 81442, Saudi Arabia

**Keywords:** gold nanoparticles, antibiotic resistance, vancomycin, metallic nanoparticles, transmission electron microscopy

## Abstract

The remarkable rise of antibiotic resistance among pathogenic bacteria poses a significant threat to human health. Nanoparticles (NPs) have recently emerged as novel strategies for conquering fatal bacterial diseases. Furthermore, antibiotic-functionalized metallic NPs represent a viable nano-platform for combating bacterial resistance. In this study, we present the use of vancomycin-functionalized gold nanoparticles (V-GNPs) to battle pathogenic bacterial strains. A facile one-pot method was adopted to synthesize vancomycin-loaded GNPs in which the reducing properties of vancomycin were exploited to produce V-GNPs from gold ions. UV–Visible spectroscopy verified the production of V-GNPs via the existence of a surface plasmon resonance peak at 524 nm, whereas transmission electron microscopy depicted a size of ~24 nm. Further, dynamic light scattering (DLS) estimated the hydrodynamic diameter as 77 nm. The stability of V-GNPs was investigated using zeta-potential measurements, and the zeta potential of V-GNPs was found to be −18 mV. Fourier transform infrared spectroscopy confirmed the efficient loading of vancomycin onto GNP surfaces; however, the loading efficiency of vancomycin onto V-GNPs was 86.2%. Finally, in vitro antibacterial studies revealed that V-GNPs were much more effective, even at lower concentrations, than pure vancomycin. The observed antibacterial activities of V-GNPs were 1.4-, 1.6-, 1.8-, and 1.6-fold higher against Gram-negative *Escherichia coli*, *Klebsiella oxytoca*, and *Pseudomonas aeruginosa* and Gram-positive *Staphylococcus aureus*, respectively, compared to pure vancomycin. Collectively, V-GNPs represented a more viable alternative to pure vancomycin, even at a lower antibiotic dose, in conquering pathogenic bacteria.

## 1. Introduction

Bacterial resistance to existing antibiotics has risen, owing to the overuse of antibacterial drugs. Vancomycin is a glycopeptide antibiotic that is commonly used for the treatment of severe but susceptible bacterial infections such as methicillin-resistant *Staphylococcus aureus* (MRSA) [1]. Generally, the antibacterial activity of vancomycin is attributed to its ability to inhibit bacterial cell wall biosynthesis via binding to a specific C-terminal sequence, d-Ala–d-Ala, in the peptidoglycan pentapeptide [2]. Nevertheless, the widespread use of vancomycin has resulted in the emergence of vancomycin-resistant bacterial strains [3,4]. Furthermore, in contrast to various antibiotics, such as penicillin, cephalosporin, and fluoroquinolones, the efficacy of vancomycin against many Gram-negative bacteria is substantially compromised by its inability to diffuse through porins [5]: proteins present in the outer membranes of Gram-negative bacteria that form channels to allow the entry of antibiotics due to the fact of its large size and high molecular mass. Accordingly, alternative approaches for conveying the antibiotic package inside the bacterial cell is necessary such as applying nanotechnology-based techniques or attachment of polycationic peptide [6]. 

Nanotechnology offers a cutting-edge discipline that has the potential to treat infections in novel ways using nanoparticles. The easy-to-manipulate characteristics of nanoparticles, such as shape, size, and surface chemistry, make them versatile platforms in fighting bacterial infections [7,8]. In addition, by virtue of their high drug loading efficiency and greater capacity to permeate biological membranes, nanoparticles represent excellent candidates for antimicrobial agents’ delivery at the site of infection [9,10]. Furthermore, the ability of nanoparticles to interact with bacterial cellular systems substantially participates in the enhanced therapeutic efficacy of the treatment [11]. In addition, nanoparticles have been considered as plausible tools for the diagnosis of various clinical diseases [12]. Moreover, the success of nanoparticle-based therapeutic strategy depends on their safety assessment [13], large-scale production, and applicability for clinical trials [14]. A wide arsenal of metallic nanoparticles, including gold, silver, zinc oxide, copper oxide, aluminum oxide, and titanium dioxide nanoparticles, are currently being evaluated, alone or conjugated with antibiotics, for their potential in fighting bacterial infections [15,16,17]. Among them, gold nanoparticles (GNPs) have received immense attention because of their lower toxicity, optical properties and, most importantly, ease of surface functionalization [18,19].

Recently, a mounting body of literature has emphasized the feasibility of utilizing GNPs as carriers for antimicrobial agents [19,20,21,22]. GNPs efficiently enable the delivery of relatively higher drug concentrations at the site of infection while, simultaneously, minimizing the drug’s toxicity [23]. In addition, GNPs were acknowledged for their efficiency to sustain antibiotic release over a prolonged period of time, resulting in an enhanced antibiotic efficacy [24]. 

Beside enhanced delivery, many reports have demonstrated that conjugating GNPs with antibiotics tends to enhance the antibacterial activities of antibiotics while mitigating side effects via reducing the need for high antibiotic doses [25,26,27]. For instant, Lee et al. [27] demonstrated that combinations of GNPs with ciprofloxacin or cefotaxime exerted a synergistic effect against all *Salmonella* species via inducing bacterial apoptosis-like death. Similarly, Fuller et al. [26] reported that conjugating the antibiotic colistin to GNPs significantly augmented the antibacterial activity of colistin against *Escherichia coli* (*E. coli*) as manifested by a six-fold decrease in the minimum inhibitory concentration (MIC), compared to colistin alone. Most importantly, antibiotic conjugation to GNPs has been reported to reverse antibacterial resistance to various antibiotics [25,28]. Brown et al. [29] demonstrated that conjugating ampicillin to the surface of GNPs significantly enhanced the antibacterial activity against methicillin-resistant *Staphylococcus aureus*, *Enterobacter aerogenes*, and *Pseudomonas aeruginosa*. In the same context, Shaikh et al. [30] revealed that conjugating cefotaxime (CTX) to GNPs efficiently restored the antibacterial activity of cefotaxime against resistant bacterial pathogens: *E. coli* and *Klebsiella pneumonia.* Collectively, antibiotic-functionalized GNPs might hold great promise for delivering the loaded drug at lower doses with improved efficacy and, eventually, reducing drug-related dose-dependent side effects.

In the current study, therefore, we challenged the efficacy of GNPs to potentiate the antibacterial efficacy of the antibiotic vancomycin against various bacterial strains. A facile one-pot method was adopted to functionalize GNPs with vancomycin in which vancomycin had dual roles as a reducing agent and a capping agent. The physicochemical properties of vancomycin-functionalized gold nanoparticles (V-GNPs) were evaluated using UV–Visible spectrophotometry, dynamic light scattering technique, transmission electron microscopy, and Fourier transform infrared spectroscopy (FTIR). The antibacterial potential of V-GNPs was assessed against both G –ve and G +ve bacteria and was compared to that of pure vancomycin. Our results emphasize the superior antibacterial activity of V-GNPs against tested bacterial strains compared to pure drug.

## 2. Materials and Methods

### 2.1. Materials

Vancomycin, gold (III) chloride trihydrate (HAuCl_4_ · 3H_2_O) and phosphate buffer were purchased from Sigma–Aldrich (St. Louis, MO, USA). Müeller Hinton (MH) broth agar and Luria–Bertani (LB) broth were procured from HiMedia (Mumbai, India). All other chemicals and solvents were of analytical grade.

### 2.2. Bacterial Strains, Cell Lines, and Cultivation Conditions

The Gram-negative bacterial strains Escherichia coli (ATCC 25923), Klebsiella oxytoca (ATCC 43165), and Pseudomonas aeruginosa (NCIM 2036) and the Gram-positive bacteria Staphylococcus aureus (ATCC 14222) were obtained from the Indian National Chemical Laboratory (Pune, Maharashtra, India). Each bacterial strain was cultivated and maintained at 37 °C on MH agar media. Standardized suspensions of the tested strains (equivalent to the 0.5 McFarland) were prepared from overnight cultures in TSB and swabbed over the surface of Müeller–Hinton agar plates (Himedia, Mumbai, India).

### 2.3. Vancomycin-Mediated Synthesis of GNPs

For the preparation of V-GNPs, varying amounts of vancomycin (0.25, 0.50, 0.75, or 1 mg/mL) were added to a 3 mL reaction mixture, containing 1 mM gold (III) chloride trihydrate in a 50 mM phosphate buffer (pH of 7.4). The reaction mixture was then incubated for 48 h at 30, 40, 50, or 60 °C, respectively. A similar reaction was conducted without the use of vancomycin that served as a control. Upon completion of the reaction, as indicated by color change from green to ruby red, the synthesized V-GNPs were recovered by centrifuging the reaction mixture at 30,000× *g* for 30 min. The collected NPs were then rinsed twice with Milli-Q water and finally treated with 50% *v*/*v* ethanol to remove unattached materials. 

### 2.4. Characterization of the Synthesized V-GNPs

#### 2.4.1. UV–Vis Spectroscopy

The synthesis of GNPs from gold salts was confirmed by a Shimadzu UV-1601 spectrophotometer (Shimadzu Corporation, Tokyo, Japan), which was utilized to the acquire UV–Visible spectra of the synthesized V-GNPs at a resolution of 1 nm within the range of 200–800 nm.

#### 2.4.2. Dynamic Light Scattering (DLS) and Zeta Potential Analysis

A dynamic light scattering particle size analyzer (Zetasizer Nano-ZS, Malvern Instrument Ltd., Worcestershire, UK) was used to determine the average particle size of V-GNPs. Prior subjecting to DLS, the samples were sonicated for 1 min and were filtered via 0.45 µm membrane syringe filters. The samples were then placed inside a 1.5 mL DTS0112-low-volume disposable sized cuvette. The average particle size was determined as the mean of three independent experiments. 

In order to identify the kind of charge present onto the surface of the synthesized V-GNPs, a Malvern Zetasizer Nano-ZS (Malvern Instrument Ltd., Worcestershire, UK) was adopted to measure the zeta potential of the prepared NPs. Sample were prepared in a similar way as for the DLS measurement, and DTS1070 disposable cuvettes were used for zeta potential analysis. Three independent zeta-potential analyses of the sample were performed under identical experimental conditions, and the mean was considered the final zeta-potential value. 

#### 2.4.3. Transmission Electron Microscopy (TEM)

The homogeneity of the synthesized V-GNPs was characterized using TEM (Tecnai G2 Spirit outfitted with a Bio-TWIN CCD camera, Hillsboro, OR, USA) operated at an accelerating voltage of 80 kV. V-GNPs samples were prepared on a carbon-coated copper grid and allowed to air dry. Any remaining solution was removed with filter paper prior to TEM analysis. 

#### 2.4.4. Fourier Transform Infrared (FTIR) Spectroscopy 

Fourier transform infrared (FTIR) spectroscopy was conducted to assess the conformational changes upon the loading of vancomycin onto the surface of GNPs. The KBr pellet method [31,32] was adopted to acquire FTIR spectra with a Shimadzu FTIR-8201 spectrometer (Shimadzu, Tokyo, Japan) at a resolution of 4 cm^−1^ in the range of 4000–400 cm^−1^.

### 2.5. Loading Efficiency of V-GNPs

The loading efficiency of V-GNPs was estimated as previously described [33]. Briefly, V-GNPs were centrifuged at 30,000× *g* for 30 min. Free vancomycin in the supernatant was then quantified spectrophotometrically at λmax 278 nm. The concentration of vancomycin in the supernatant was calculated from a pre-constructed calibration curve of vancomycin within a concentration range of 100–500 µg/mL. The percentage loading efficiency of vancomycin was finally estimated using the following equation:$$\mathrm{Loading}\;\mathrm{efficiency}\;(\%)=\frac{Total\;amount\;of\;drug-free\;drug\;in\;the\;supernatant}{Total\;amount\;of\;drug}\times 100$$

### 2.6. Antibacterial Activity Evaluation 

#### 2.6.1. Qualitative Evaluation of Antibacterial Activity 

In order to assess the efficacy of synthesized V-GNPs, compared to pure vancomycin, the agar well diffusion methodology was used [34]. Briefly, Mueller–Hinton agar plates were inoculated with a standardized inoculum of the test microorganisms, namely, *Escherichia coli, Klebsiella oxytoca, Pseudomonas aeruginosa*, and *Staphylococcus aureus*. Then, using a sterilized well cutter, a hole (6 mm in diameter) was punched aseptically, and a volume of 100 µL of V-GNPs (equivalent to 21.5 µg vancomycin/well), pure vancomycin (50 µg/well), and phosphate buffer saline (PBS; control) were poured into the wells. The agar plates were then incubated under appropriate conditions overnight at 37 °C. All experiments were performed in triplicates, and the diameter of the inhibitory zone was determined as the mean ± standard deviation.

#### 2.6.2. Minimal Inhibitory Concentration (MIC) Determination

The MIC method was adopted to assess the in vitro antibacterial activity of either V-GNPs or pure vancomycin against different bacterial strains. To determine the MIC, the broth microdilution method was conducted as described previously [35]. Briefly, aliquots of V-GNPs and pure vancomycin were serially diluted in 96-well microtiter plates using LB broth medium to obtain concentrations ranging from 6.25 to 200 µg/mL. The tested bacterial strains were inoculated in LB broth and incubated overnight at 37 °C. The resultant suspension was diluted with sterile saline to a turbidity of 0.5 McFarland standard, equivalent to 1 × 10^8^ CFU/mL. Then, 10 µL aliquots of the standard suspensions were added to all vancomycin dilutions, and the plate was incubated at 37 °C for 20 h. The MIC was determined by monitoring the lowest drug concentration that prevented the growth.

### 2.7. Statistical Analysis

Each sample was tested in triplicates, and the results were expressed as the mean ± standard deviation. The findings were analyzed via one-way analysis of variance (ANOVA) using GraphPad Prism (Graph Pad Software, Version 4.02, San Diego, CA, USA). A *p* < 0.05 implies significant difference.

## 3. Results and Discussion

The emergence of vancomycin-resistant bacterial strains [3,4] has raised a serious clinical concern, and alternatives pragmatic solutions are urgently needed. In the present study, GNPs were used as a tool to deliver and augment the activity of vancomycin against different pathogenic strains. Vancomycin mediated the synthesis of GNPs and successfully loaded with 86.2% efficiency onto the GNPs (Section 3.1 and Section 3.3). Characterization showed efficient GNP formation by UV–Visible spectroscopy, appropriate nano-size by DLS and TEM, good stability by zeta potential, and successful loading of vancomycin on GNPs by FTIR (Section 3.2). Antibacterial analysis (Section 3.4 and Section 3.5) showed significant increase in the activity of vancomycin once loaded onto GNPs as compared to pure vancomycin. 

### 3.1. Vancomycin-Mediated Synthesis of Vancomycin-Loaded Gold Nanoparticles 

Despite the fact that GNPs themselves are generally considered to be biologically inert and do not possess antibacterial activity [22], recent reports have revealed an antibacterial action for “pristine” GNPs against different bacterial strains [36,37]. Nevertheless, in most cases, the observed antibacterial action might be attributed to other chemicals used in the synthesis process rather than GNPs themselves [19,38]. This is a critical factor to be considered upon evaluating the origin of the antimicrobial action. In the current study, therefore, we adopted a one-pot method for the synthesis of V-GNPs in which vancomycin itself played dual roles as a reducing agent and a capping agent (Figure 1) to nullify the possible tangling effect of the use of external reducing/capping agents on the antibacterial activity of the synthesized GNPs. Furthermore, this method offered the advantage of permitting the synthesis of GNPs and concomitant loading of the drug (vancomycin) onto the surface of synthesized GNPs from the same reaction mixture, without the need of using an external conjugating agent and/or multi-step reaction. 

Many reports have emphasized the impact of various reaction conditions, such as metal salt concentration, reducing/capping agent concentrations, temperature, pH, and incubation time, on the morphology (i.e., shape and size) of the synthesized gold nanoparticles [39,40]. Accordingly, in this study, preliminary experiments were conducted to optimize the reaction conditions for the synthesis of GNPs using varying concentrations of vancomycin (0.25, 0.50, 0.75, or 1 mg) and temperature conditions (30, 40, 50, or 60 °C). The results of these experiments revealed that the size, shape, monodispersity, and overall stability of the V-GNPs were mostly influenced by the vancomycin concentration and reaction temperature adopted (data not shown). In addition, among the different reaction conditions adopted, GNPs prepared at a vancomycin concentration of 250 µg/mL, at a temperature of 40 °C, and a pH of 7.4 showed appropriate physiochemical properties, and were selected for further investigations.

### 3.2. Characterization of Synthesized V-GNPs

#### 3.2.1. UV–Visible Spectroscopic Analysis

The color change of the gold salt solution (1 mM HAuCl_4_ in phosphate buffer; pH 7.4) from pale yellow to ruby red after incubation with 250 µg/mL vancomycin antibiotic (Figure 2A), provides a visual indication for the formation of vancomycin-loaded gold nanoparticles. However, to further confirm the efficient synthesis of V-GNPs, the UV–Visible spectra of the synthesized V-GNPs were recorded. It is well known that noble metal nanoparticles display unique optical properties on account of their surface plasmon resonance (SPR) [41]. In this study, the presence of a sharp absorption peak at 524 nm, corresponding to the plasmon band of GNPs (Figure 2B), strongly verified the successful synthesis of GNPs. Furthermore, the existence of an absorption peak at 278 nm, corresponding to vancomycin, validated the adhesion/loading of vancomycin molecules onto the surface of the synthesized GNPs (Figure 2B). Similarly, many reports have emphasized the existence of an antibiotic’s prominent absorption peak along with characteristic plasmon peak of GNPs in antibiotic-functionalized gold nanoparticles [30,40]. Our study confirmed that vancomycin utilized in the precursor solution might function as a potent reducing agent, leading to the reduction of gold salts in GNPs. Additionally, vancomycin might stabilize the prepared V-GNPs by being adsorbed onto the NPs’ surface, preventing them from aggregating.

#### 3.2.2. Morphology and Diameter Distribution

The morphology and diameter distributions of V-GNPs were investigated using the TEM analysis. The TEM micrographs (Figure 3) depicts that the synthesized V-GNPs were spherical in shape, homogeneous, and monodispersed with an average size of ~24 nm. Most importantly, no signs of aggregation or agglomeration were observed in the TEM micrographs, indicating the efficacy of vancomycin as a capping/stabilizing agent.

#### 3.2.3. Dynamic Light Scattering and Zeta-Potential Analysis

The size of the V-GNPs measured by DLS was found to be 77 nm (Appendix A). The relatively larger size of the V-GNPs estimated by DLS, compared to that of TEM, might be attributed to the presence of dispersant in DLS. Generally, in TEM analysis, particle size estimation is conducted in the dry state; TEM determines the exact size of each particle, excluding the contribution of the surrounding solvent layer of dispersant. On the other hand, the DLS approach determines the size as a hydrodynamic diameter (hydrated state); thereby, as a result of the solvent effect in the hydrated state, the particles will have a greater hydrodynamic volume [42].

Generally, zeta potential is considered a key determinant of the colloidal stability of nanoparticles. In this study, the estimated zeta potential of the synthesized V-GNPs was −18 mV, indicating the good stability of the GNPs (Appendix A). The data portrayed by zeta potential findings provide information about the exposure or shielding of charged moieties, ionization, adsorption, and distribution of the nanoparticle [39]. It is worth noting that in zeta-potential analysis, only repulsive electrostatic forces, such as Van der Waals forces, are assessed; accordingly, we did not rely solely on zeta-potential data in ensuring the colloid stability of the prepared NPs. Instead, the stability of the synthesized V-GNPs was re-examined after 5 months at room temperature. No further aggregation was detected, indicating that the synthesized V-GNPs were highly stable in nature.

#### 3.2.4. Fourier Transform Infrared Spectroscopy 

FTIR is used conduct semi-quantitative measurements on a wide variety of organic molecules (including adsorbed/immobilized compounds). The benefits of FTIR include immediate determination of adsorbed/immobilized compounds that are difficult to quantify by UV–Visible absorption. In fact, different chemical structures (molecules) create spectral information with distinct characteristics that could be detected with the help of FTIR. In the present study, FTIR spectroscopy was employed to confirm the loading of vancomycin onto the surface of synthesized GNPs, since FTIR analysis could fingerprint chemical components in the synthesized V-GNPs. Figure 4 shows the main peaks in the FTIR spectra of pure vancomycin as well as the V-GNPs produced. The obtained FTIR spectrum of vancomycin exhibited distinctive peaks at 3401.02 cm^−1^ for O–H (stretching), 1635.35 cm^−1^ for C=O (stretching), 1383.36 cm^−1^ for C=C (stretching), and 1065.49 cm^−1^ for the phenolic hydroxyl groups (Figure 4). Of interest, the FTIR spectrum of V-GNPs (Figure 4) exhibited a similar pattern to that of unloaded vancomycin (pure form) with minor shifts in the obtained peaks. Nevertheless, no new additional peaks were observed in the fingerprint region of the spectrum of the V-GNPs, demonstrating the presence of vancomycin in V-GNPs which, in turn, confirmed the efficient loading of vancomycin onto the surface of GNPs. Recently, a similar FTIR approach for the characterization of ceftriaxone-loaded GNPs was used by Alshammari et al. [21].

### 3.3. Percentage Loading Efficiency of V-GNPs

The estimation of drug loading/capping efficiency is an essential parameter for the characterization of nanoparticles. The loading/capping efficiency of vancomycin onto GNPs, as measured by the percentage of the drug effectively capped/loaded onto the surface of the nanoparticle, was determined to be 86.2% (from Equation (1)). In this study, 215.5 µg out of 250 µg vancomycin added to the reaction mixture was loaded onto the surface of GNPs. This finding strongly reflects the efficient loading of vancomycin onto the surface of V-GNPs and eliminates the possibility of considerable drug loss during our suggested preparation conditions.

### 3.4. Antibacterial Activity Analysis of V-GNPs

The antibacterial activities of V-GNPs and pure vancomycin were confirmed by testing them against Gram-negative (i.e., *Escherichia coli*, *Klebsiella oxytoca*, and *Pseudomonas aeruginosa*) and Gram-positive (i.e., *Staphylococcus aureus*) bacterial strains. The findings demonstrated that both pure vancomycin and V-GNPs diffused into the agar and inhibited the bacterial growth significantly (as shown in Table 1); however, the concentration of vancomycin in V-GNPs was 21.5 µg/well; that is less than half the concentration of pure vancomycin (50 µg/well). Thus, the results indicate that a relatively smaller quantity of V-GNPs was more effective than pure vancomycin. Table 1 shows the results as a zone of inhibition.

### 3.5. Determination of Minimal Inhibitory Concentration (MIC) of VM and V-GNPs

The MIC_50_ is the concentration of pure vancomycin and V-GNPs that inhibits 50% of the bacterial population. In the performed study, pure vancomycin and V-GNPs were tested against Gram-negative and Gram-positive bacterial strains (Figure 5). The quantified MIC_50_ concentrations for vancomycin and V-GNPs against *Escherichia coli* were 139.61 and 93.44 µg/mL, respectively (Figure 5A). However, the MIC_50_ values for vancomycin and V-GNPs against *Klebsiella oxytoca* and *Pseudomonas aeruginosa* were estimated as 117.7 and 70.84 µg/mL and 113.2 and 60.65 µg/mL, respectively (Figure 5B,C). Both vancomycin and V-GNPs were most potent against *Staphylococcus aureus* with an MIC_50_ value of 48.18 and 30.63 µg/mL, respectively (Figure 5D). These findings demonstrate that vancomycin loaded on V-GNPs increased the antibacterial potential of vancomycin by 1.4-, 1.6-, 1.8-, and 1.6-fold against *Escherichia coli*, *Klebsiella oxytoca*, *Pseudomonas aeruginosa*, and *Staphylococcus aureus*, respectively.

The antibacterial results revealed that the V-GNPs were significantly more effective at lower doses than pure vancomycin against the bacterial strains tested, plausibly due to the fact of their loading onto GNPs that lowered the effective antibiotic dosage and increased the potency. GNPs not only acted as an effective carrier for V, it might have exerted an additional antibacterial potential by collapsing the membrane potential and inhibiting bacterial ATPase activities [43]. However, vancomycin, once carried by the V-GNPs in appropriate concentration, will act by attaching to nascent cell wall mucopeptides with the sequence d-Ala–d-Ala, thereby weakening the cell wall and, as a result, the transpeptidase might be inhibited, which could prevent further elongation and cross-linking of the peptidoglycan matrix [44].

Thus, it can be inferred from our study that in V-GNPs both the antibiotic vancomycin and GNPs are exerting the antibacterial potential synergistically. It has been observed that GNPs alone in a size range of 20–40 nm showed potent antibacterial activity with MICs of 2.96, 3.98, and 3.3 μg/mL against *E. coli*, *S. aureus*, and *K. pneumonia*, respectively [45]. It is noteworthy to mention that GNPs are one of the most preferred inorganic nanoparticles for drug administration due to the fact of their biocompatibility, non-cytotoxicity, and excellent physiochemical characteristics [46,47]. Importantly, the surface exposure of antibiotic (vancomycin) when linked to GNPs might have retained its antibacterial action [24]. This is in line with previous findings that ceftriaxone, cefotaxime, and ampicillin retain efficacy following conjugation to GNPs [21,29,30]. Cefotaxime–GNPs showed MIC values of 1.009 and 2.018 mg/L against drug-resistant *E. coli* and *K. pneumonia*, respectively [30]. Whereas the MICs of ceftriaxone–GNPs against *E. coli*, *S. aureus*, and *K. pneumoniae* were calculated as 1.39, 1.6, and 0.9 µg/mL, respectively [18]. Importantly, all the antibiotic-loaded GNPs [21,29,30] showed better potential than pure antibiotic, which is in agreement with our findings.

The hypothesis on the mechanistic aspects of the antibacterial potential enhancement of vancomycin loaded onto V-GNPs is in line with the earlier findings in [21,30,48]. Firstly, the effective delivered concentration of vancomycin was elevated due to attachment to the surface of GNPs. It might be due to the extensive surface area, excellent permeability to the biological membrane, and higher uptake by the bacterial cell [49]. Secondly, GNPs might have interacted with bacterial lipid bilayers and lipopolysaccharides, triggering membrane fluidization and membrane disruption. In fact, the ions liberated by the nanoparticle infiltrate the cytosol through the pores of the membrane, and the nanoparticles could also affect membrane proteins (e.g., efflux pumps) and enzymes (e.g., oxidoreductase family), generating metabolic inefficiency [43]. This leads to structural collapse, poor surface adhesion, metabolic obstruction and, ultimately, bacterial cell death [43]. Therefore, the performed study not only portrayed the enhanced antibacterial effect of the vancomycin after loading on GNPs but also rationalized the synergistic effect of the vancomycin and GNPs.

In our investigation, GNPs significantly increased the potency of vancomycin, but its destiny in the human body, mechanistic pathway, and toxicity are still unexplored. V-GNPs are now being studied in vivo and in vitro to determine their specific mode of action, toxicity, and lethal dosage. Our team hopes to translate these preliminary findings into applicable effective nano-formulations against bacterial pathogens.

## 4. Conclusions

The current work demonstrated a method for synthesizing gold nanoparticles using vancomycin as a reducing and capping material. Additionally, synthesized V-GNPs were proved to be an efficient delivery system of vancomycin to the tested bacterial strains. A comparison of pure vancomycin and V-GNPs revealed that vancomycin could become extremely potent if loaded onto GNPs. As vancomycin resistance is emerging at a fast pace, the conversion of available vancomycin to more potent formulations is a better strategy than spending large amounts of resources and time on developing newer antibiotics. Importantly, the current study approach paves the way for the concurrent synthesis and delivery of several antibiotics employing GNPs, therefore resolving the issue of increasing resistance. However, before establishing a convincing conclusion about the applicability of synthesized vancomycin nano-formulations, in vivo investigations to determine the destiny and toxicity of V-GNPs are required. In fact, large-scale production and proper safety profiling is still not well established in the case of nano-formulations. Nevertheless, the preliminary findings of this study could serve as a baseline for developing appropriate antibacterial nano-formulations.

## Figures and Tables

**Figure 1 materials-15-01108-f001:**
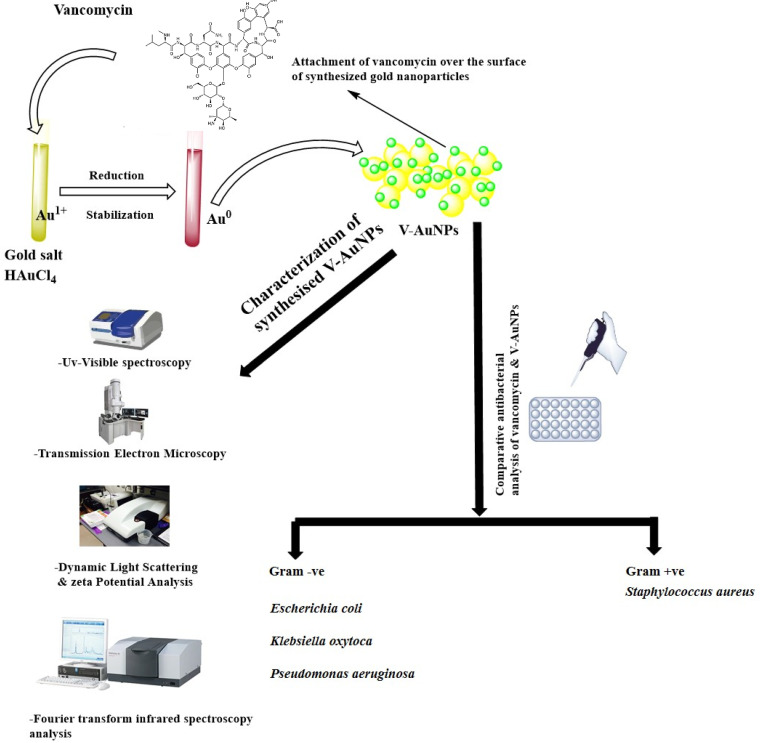
Schematic representation of vancomycin-mediated synthesis of gold nanoparticles (V-GNPs), their characterization, and antibacterial testing.

**Figure 2 materials-15-01108-f002:**
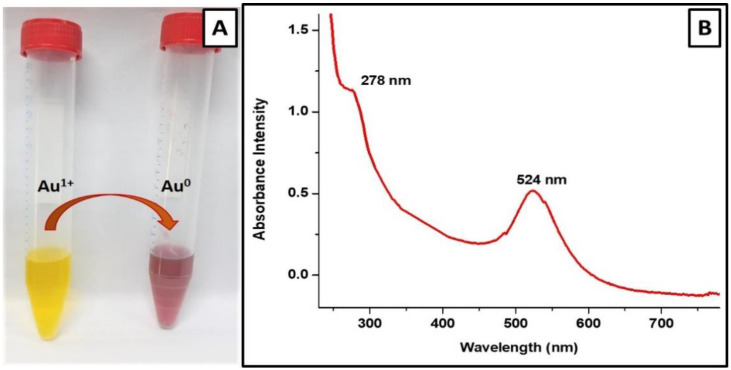
Characterization of V-GNPs: (**A**) color change from light yellow to ruby red resulted from SPR; (**B**) UV–Visible spectra (SPR band at 524 nm).

**Figure 3 materials-15-01108-f003:**
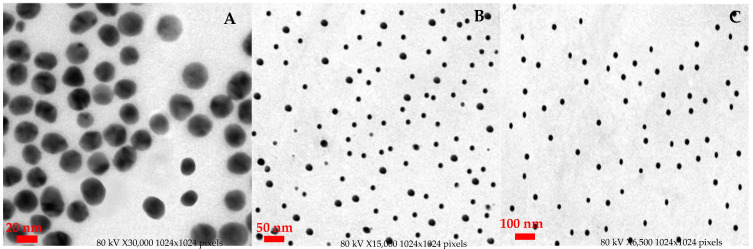
TEM of V-GNPs representing spherical monodispersed particles at the scale of (**A**) 20; (**B**) 50; (**C**) 100 nm with an average size of ~24 nm.

**Figure 4 materials-15-01108-f004:**
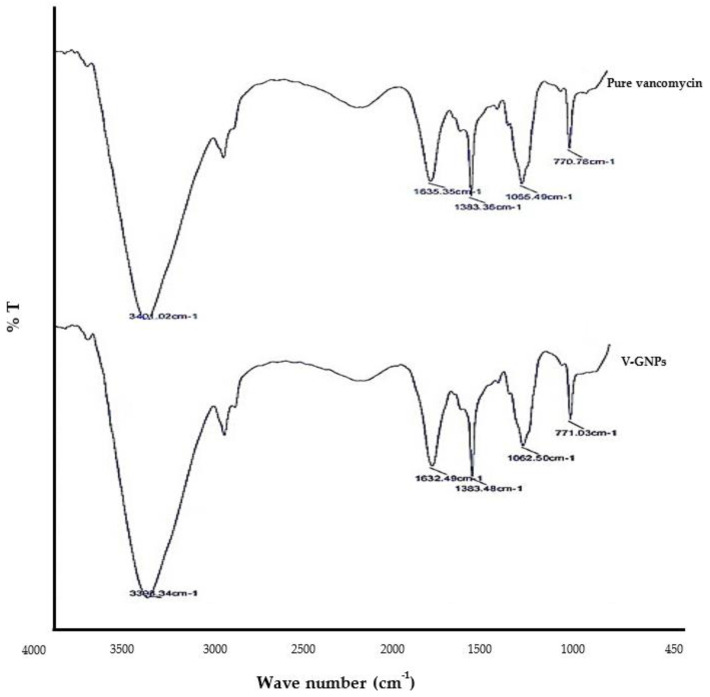
FTIR spectra of pure vancomycin and vancomycin-loaded gold nanoparticles (V-GNPs).

**Figure 5 materials-15-01108-f005:**
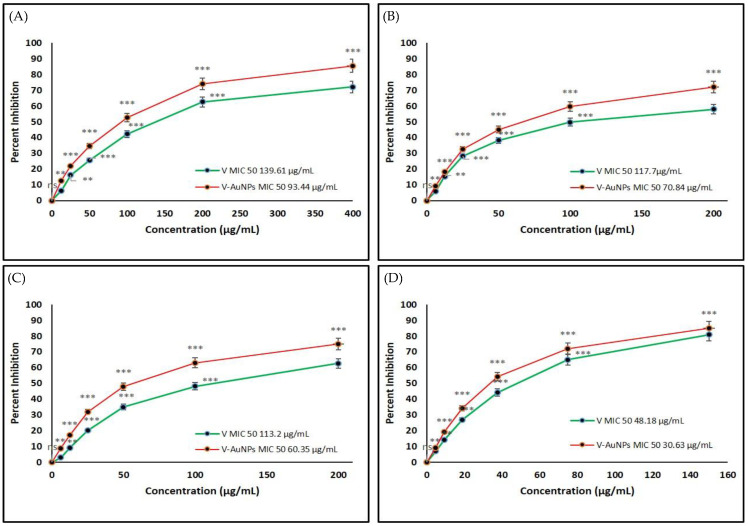
Determination of the minimum inhibitory concentration (MIC) of vancomycin (V) and V-GNPs against (**A**) *Escherichia coli;* (**B**) *Klebsiella oxytoca*; (**C**) *Pseudomonas aeruginosa*; (**D**) *Staphylococcus aureus.* The experiment was repeated in triplicate, and the data shown are the means ± standard errors. Significantly different from control at ** *p <* 0.01*,* *** *p* < 0.001; non-significant from the control at ^ns^ *p* > 0.05.

**Table 1 materials-15-01108-t001:** Antibacterial analysis of pure vancomycin and vancomycin loaded gold nanoparticles (V-GNPs) against different bacterial strains.

Zone of Inhibition (mm)				
Sample	*Escherichia* *coli*	*Klebsiella* *oxytoca*	*Pseudomonas aeruginosa*	*Staphylococcus aureus*
Control	NZ ^ns^	NZ ^ns^	NZ ^ns^	NZ ^ns^
Vancomycin	22 ± 0.10 ***	16 ± 0.04 **	18 ± 0.01 ***	34 ± 0.16 ***
V-GNPs	15 ± 0.14 ***	11 ± 0.02 **	14 ± 0.12 ***	24 ± 0.14 ***

The values are the mean ± standard deviation of three independent experiments performed under identical experimental conditions. NZ—No zone of inhibition; the values are the mean ± standard deviation (*n* = 3). Significantly different from the control at ** *p <* 0.01*,* *** *p* < 0.001; non-significant from the control at ^ns^ *p* > 0.05.

## Data Availability

Not applicable.

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
