# Peer review of "Enhancement of Vancomycin Potential against Pathogenic Bacterial Strains via Gold Nano-Formulations: A Nano-Antibiotic Approach"

_materials, 2022, doi:10.3390/ma15031108_

Round 1

Reviewer 1 Report

The manuscript “Enhancement of vancomycin potential against pathogenic bacterial strains via gold nanoformulations: a nano-antibiotic approach” by  Alhagbani et al. reports the synthetic methodology of antibiotic-functionalized metallic NPs and their full characterization in order to be applied as antibacterial agents for several kinds of bacteria (including gram-positive and gram-negative strains). The manuscript is well organized and presented, allowing an easy interpretation of the experimental data covering the physicochemical and microbiological aspects of the synthesized materials. I recommend their publication in Materials after minor changes, which are detailed below:

1-       Section 2.4.2. Include Z-potential in the title and briefly detail the measurement conditions. Are average from multiple runs?

2-       Section 2.6.1. Check the strain names, are not in italic letters in this section.

3-       Page 4. Numbering the equation of Loading Efficiency should be necessary.

4-       Fig 2.a- Include the image that represents the yellow dispersion in order to show the change. Fig. 2.b- Improve the quality of the plot.

5-       Fig. 3. Scale bars should be more visible, please improve them. By the way, why is not reported an average diameter (histogram) of the particles measured from the TEM images?

6-       Fig. 4 looks like raw data. Please, change the plots for others in concordance with the rest of the figures of the manuscript. How many times authors did the measurement of DLS? Please, report an average considering repetitions.

7-       Fig. 5. Put the label in the “x-axis”.

8-       About inhibition zone… is possible that authors incorporate some representative images of the Petri dishes for qualitative comparison?

Author Response

First of all, we would like to thank the honorable reviewer to take time from the busy schedule to review and improve our manuscript. All the comments have been addressed and highlighted in Yellow.

Reviewer 1:

Comment 1:      Section 2.4.2. Include Z-potential in the title and briefly detail the measurement conditions. Are average from multiple runs?

Authors response: According to the suggestion of the honorable reviewer, Zeta potential and brief description of the conditions has been duly added in the section 2.4.2. Yes, the average value of three runs has been considered during the study and a statement regarding this has been added in the section 2.4.2 of the revised MS.

Comment 2:      Section 2.6.1. Check the strain names, are not in italic letters in this section.

Authors response: Thanks for pointing out the typographical mistake, the strain names have been rewritten into italics.

Comment 3:       Page 4. Numbering the equation of Loading Efficiency should be necessary.

Authors response: As per the reviewer's suggestions, equation number (equation 1) has been added to the loading efficiency equation in the revised MS.

Comment 4:       Fig 2.a- Include the image that represents the yellow dispersion in order to show the change. Fig. 2.b- Improve the quality of the plot.

Authors response: As per the suggestion of the honorable reviewer, the yellow dispersion image has been duly added in the revised figure 2a, and figure 2b has been improved.

Comment 5:       Fig. 3. Scale bars should be more visible, please improve them. By the way, why is not reported an average diameter (histogram) of the particles measured from the TEM images?

Authors response: As per the suggestion of the honorable reviewer, the figure 3 has been improved to make scale bars more visible. However, the average diameter of the nanoparticles obtained via TEM analysis has been mentioned in the manuscript as 24 nm.

Comment 6:      Fig. 4 looks like raw data. Please, change the plots for others in concordance with the rest of the manuscript's figures. How many times do authors do the measurement of DLS? Please, report an average considering repetitions.

Authors response: As  per the suggestion of the honorable reviewer, the Figure 4 has been improved and results are average of three runs that is duly mentioned in the methology section of the revised MS.

Comment 7:       Fig. 5. Put the label in the “x-axis”.

Authors response: As suggested by the reviewer missing X-axis values are now incorporated in Figure 5.

Comment 8:       About inhibition zone… is possible that authors incorporate some representative images of the Petri dishes for qualitative comparison?

Authors response: With all due respect to the honorable reviewer, we would like to state that in the present study qualitative evaluation by zone of inhibition was used only for preliminary analysis. Our main focus was on MIC quantitative determination similar to our recent study [1]. We don’t have all the figures right now, however, two of the sample figures are attached for your kind perusal. In fact, we have to repeat the experiment to obtain the figures of all the tested strain. In the current scenario, we will be thankful to the author if he consider the results without inclusion of images.

[1] Alshammari, F. , Alshammari, B. , Moin, A. , Alamri, A. , Hagbani, T. Al , Alobaida, A. , Baker, A. , Khan, S. , and Rizvi, S. M. D. Ceftriaxone mediated synthesized gold nanoparticles: A nano-therapeutic tool to target bacterial resistance. Pharmaceutics 2021, 13.

Reviewer 2 Report

The paper “Enhancement of vancomycin potential against pathogenic bacterial strains via gold nanoformulations: a nano-antibiotic approach” by Turki Alhagbani, Hemant Yadav, Afrasim Moin, Amr Selim Abu Lila, Khalid Mehmood, Farhan Alshammari, Salman Khan, El-Sayed Khafagy, Talib Hussain, Syed Mohd Danish Rizvi, Marwa H. Abdallaha describes activities of some interest. Gold nano particles are combined with antibiotic vancomycin to overcome drug resistance. Problems already arose with vancomycin in the end of the 1980 years, therefore it was substituted by linezolid. 
The basic idea of the paper is to coordinate vancomycin at gold nano particles. Gold(III)chloride was allowed to react in a one-pot method with vancomycin. The reaction product was thoroughly characterized by different spectroscopic and other physical methods. Thus, a characteristic plasmon absorption band of gold particles was found. The average radius of them was determined by TEM, whereas the hydrodynamic radius in solution was estimated by DLS. The antibacterial activities were tested against gram-negative bacteria. The efficiency  of vancomycin and V-GNPs was compared. A better insight would be obtained if identical amounts of vancomycin in both samples are applied. 
A constraint of this strategy could be the limited stability of the nano particles. 
A more straightforward improvement of this antibiotic was proposed in the following paper which should be cited: 
Florian Umstätter, Cornelius Domhan, Tobias Hertlein, Knut Ohlsen, Eric Mühlberg, Christian Kleist, Stefan Zimmermann, Barbro Beijer, Karel D. Klika, Uwe Haberkorn, Walter Mier, Philipp Uhl, Vancomycin Resistance Is Overcome by Conjugation of Polycationic Peptides; Angew. Chem. Int. Ed. Engl. 2020 Jun 2; 59 (23): 8823-8827; doi: 10.1002/anie.202002727. Epub 2020 Apr 21.

Author Response

First of all, we would like to thank the honorable reviewer to take time from the busy schedule to review and improve our manuscript. All the comments have been addressed and highlighted in Yellow.

Reviewer 2:

Comment 1: A better insight would be obtained if identical amounts of vancomycin in both samples are applied. 

Authors response: We appreciate the suggestion of the honorable reviewer. With all due respect to the reviewer, we would like to state that our prime focus was to calculate the MIC qualitatively. We used zone of inhibition for preliminary qualitative analysis purpose only. However, we will keep the valuable suggestion noted for our next article in future.

Comment 2: A constraint of this strategy could be the limited stability of the nano particles. 

Authors response: We appreciate the concern of the honorable reviewer, we have done zeta potential to check the stability of GNPs, and it is estimated as -18mV which suggest the good stability of GNPs. In addition, we kept GNPs at room temperature for 6 months and found it to be stable without aggregation.

Comment 3: A more straightforward improvement of this antibiotic was proposed in the following paper which should be cited: 

Florian Umstätter, Cornelius Domhan, Tobias Hertlein, Knut Ohlsen, Eric Mühlberg, Christian Kleist, Stefan Zimmermann, Barbro Beijer, Karel D. Klika, Uwe Haberkorn, Walter Mier, Philipp Uhl, Vancomycin Resistance Is Overcome by Conjugation of Polycationic Peptides; Angew. Chem. Int. Ed. Engl. 2020 Jun 2; 59 (23): 8823-8827; doi: 10.1002/anie.202002727. Epub 2020 Apr 21.

Authors response: As per the suggestion of the honorable reviewer, we have duly added the reference in the introduction section.

Reviewer 3 Report

The Abstract should be better summarized.

The authors should better mark in Introduction how nanotechnologies applied to bioactive compounds and pharmaceutics  can represent a new frontier, taking into account also safety aspects, and related references should be added as:

Zhu et al.. Nanomaterials as Promising Theranostic Tools in Nanomedicine and Their Applications in Clinical Disease Diagnosis and Treatment. Nanomaterials (Basel). 2021;11(12):3346. doi:10.3390/nano11123346

Zielińska et al. Nanotoxicology and Nanosafety: Safety-By-Design and Testing at a Glance. Int J Environ Res Public Health. 2020 Jun 28;17(13):E4657. doi: 10.3390/ijerph17134657. PMID: 32605255.

Souto et al. Nanopharmaceutics: Part II-Production Scales and Clinically Compliant Production Methods. Nanomaterials (Basel). 2020, 10(3). pii: E455. doi: 10.3390/nano10030455.   

The use of innovative methodologies such as FTIR should be better marked.

Methodologies should be described with major details, including also a graphical scheme.

Introductory lines should be inserted at the beginning of Section Results and Discussion to better describe different types of Results. Figure 1 should be improved in  a clearer way and major details added.

FTIR results should be better explain and the related Figure should be greatly improved.

Results in Figure 6 should be better described and discussed in the text.

Limits, advantages, novelty and practical applications should be described in Conclusion.

The linguistic revision of whole manuscript should be carried out.

Author Response

First of all, we would like to thank the honorable reviewer to take time from the busy schedule to review and improve our manuscript. All the comments have been addressed and highlighted in Yellow.

Reviewer 3:

Comment 1: The Abstract should be better summarized.

Authors response: As suggested by the honorable reviewer, the abstract has been summarized in a better manner with the addition of some data.

Comment 2: The authors should better mark in Introduction how nanotechnologies applied to bioactive compounds and pharmaceutics can represent a new frontier, taking into account also safety aspects, and related references should be added as:

  • Zhu et al.. Nanomaterials as Promising Theranostic Tools in Nanomedicine and Their Applications in Clinical Disease Diagnosis and Treatment. Nanomaterials (Basel). 2021;11(12):3346. doi:10.3390/nano11123346
  • ZieliÅ„ska et al. Nanotoxicology and Nanosafety: Safety-By-Design and Testing at a Glance. Int J Environ Res Public Health. 2020 Jun 28;17(13):E4657. doi: 10.3390/ijerph17134657. PMID: 32605255.
  • Souto et al. Nanopharmaceutics: Part II-Production Scales and Clinically Compliant Production Methods. Nanomaterials (Basel). 2020, 10(3). pii: E455. doi: 10.3390/nano10030455.   

Authors response: As per the suggestion of the honorable reviewer, the applied aspects of nanomaterials have been added with the addition of the suggested references.

Comment 3: The use of innovative methodologies such as FTIR should be better marked.

Authors response: As per the suggestion of the honorable reviewer, FTIR is marked with below mentioned references.

  • Tahir K, Nazir S, Li B, Khan AU, Khan ZU, Gong PY, Khan SU, Ahmad A. Nerium oleander leaves extract mediated synthesis of gold nanoparticles and its antioxidant activity. Materials Letters. 2015 Oct 1;156:198-201.
  • Sett A, Gadewar M, Sharma P, Deka M, Bora U. Green synthesis of gold nanoparticles using aqueous extract of Dillenia indica. Advances in Natural Sciences: Nanoscience and Nanotechnology. 2016 Apr 6;7(2):025005.

Comment 4: Methodologies should be described with major details, including also a graphical scheme.

Authors response: As per the suggestion of the honorable reviewer, a graphical scheme has been added as Figure 1 in the revised MS.

Comment 5: Introductory lines should be inserted at the beginning of Section Results and Discussion to better describe different types of Results. Figure 1 should be improved in  a clearer way and major details added.

Authors response: As per the suggestion of the honorable reviewer, the introductory statement has been added to the Result and Discussion section, and Figure 1 has been replaced with more detailed figure.

Comment 6: FTIR results should be better explained, and the related Figure should be greatly improved.

Authors response: As per the suggestion of the honorable reviewer, FTIR results have been duly modified with better explanation and FTIR graph has been improved.

Comment 7: Results in Figure 6 should be better described and discussed in the text.

Authors response: As suggested by the reviewer, Figure 6 results has been dicussed in a better manner in the revised MS.

Comment 8: Limits, advantages, novelty and practical applications should be described in Conclusion.

Authors response: All the suggestions of the honorable reviewer have been duly added in the conclusion section.

Comment 9: The linguistic revision of whole manuscript should be carried out.

Authors response: The entire MS has been checked thoroughly to remove linguistic error.
